# Predicting the Performance of Concurrent Systematic Random Biopsies during Image Fusion Targeted Sampling of Multi-Parametric MRI Detected Prostate Cancer. A Prospective Study (PRESET Study)

**DOI:** 10.3390/cancers14010001

**Published:** 2021-12-21

**Authors:** Saeed Alqahtani, Xinyu Zhang, Cheng Wei, Yilong Zhang, Magdalena Szewczyk-Bieda, Jennifer Wilson, Zhihong Huang, Ghulam Nabi

**Affiliations:** 1Division of Imaging Sciences and Technology, School of Medicine, Ninewells Hospital, University of Dundee, Dundee DD1 9SY, UK; szalqahtani@dundee.ac.uk (S.A.); c.wei@dundee.ac.uk (C.W.); 2School of Science and Engineering, University of Dundee, Dundee DD1 9SY, UK; y.y.zhang@dundee.ac.uk (Y.Z.); z.y.huang@dundee.ac.uk (Z.H.); 3Department of Radiological sciences, College of Applied Medical Science, Najran University, Najran 11001, Saudi Arabia; 4Division of Population Health and Genomics, School of Medicine, University of Dundee, Dundee DD1 9SY, UK; xzhang001@dundee.ac.uk; 5Department of Clinical Radiology, Ninewells Hospital, University of Dundee, Dundee DD1 9SY, UK; m.szewczykbieda@dundee.ac.uk; 6Department of Pathology, Ninewells Hospital, University of Dundee, Dundee DD1 9SY, UK; jennifer.wilson7@nhs.scot

**Keywords:** prostate cancer, magnetic resonance imaging, prostatectomy, systematic random biopsy, targeted biopsy

## Abstract

**Simple Summary:**

The study provides a predictive model by using clinical factors in selecting men who may benefit from the addition of systematic biopsies with an image fusion targeted approach. The approach is likely to improve the detection of csPCa and avoid unnecessary detection of indolent prostate cancers.

**Abstract:**

The study was aimed to develop a predictive model to identify patients who may benefit from performing systematic random biopsies (SB) in addition to targeted biopsies (TB) in men suspected of having prostate cancer. A total of 198 patients with positive pre-biopsy MRI findings and who had undergone both TB and SB were prospectively recruited into this study. The primary outcome was detection rates of clinically significant prostate cancer (csPCa) in SB and TB approaches. The secondary outcome was net clinical benefits of SB in addition to TB. A logistic regression model and nomogram construction were used to perform a multivariate analysis. The detection rate of csPCa using SB was 51.0% (101/198) compared to a rate of 56.1% (111/198) for TB, using a patient-based biopsy approach. The detection rate of csPCa was higher using a combined biopsy (64.6%; 128/198) in comparison to TB (56.1%; 111/198) alone. This was statistically significant (*p* < 0.001). Age, PSA density and PIRADS score significantly predicted the detection of csPCa by SB in addition to TB. A nomogram based on the model showed good discriminative ability (C-index; 78%). The decision analysis curve confirmed a higher net clinical benefit at an acceptable threshold.

## 1. Introduction

Recent trends and evidence support pre-biopsy MRI with selective targeting of suspected malignant lesions using MRI/ultrasound (US) and TB methods [1]. The burgeoning interest in fusion imaging has arisen to address the main limitations of SB: overdetection of clinically insignificant cancers and possibly underdetection of csPCa. A number of recent reports support the utility of pre-biopsy multi-parametric magnetic resonance imaging (mpMRI) to address the limitations of SB, and the advantage of increased csPCa detection [2,3,4]. Pre-biopsy MRI in MRI directed pathways have been reported to detect a higher number of csPCa. However, benefits of image-guided targeting performed in combination with systematic sampling or alone remains poorly defined. Drost et al., in a recent systematic review, used a mixed population (with or without image fusion targeting of suspicious areas) to answer this question; however, image fusion methods were not used in all the cases of included studies, and hence, the benefits of targeting suspicious areas with the image fusion approach, with or without addition of systematic biopsy sampling, remains unclear [5].

Image fusion approach makes use of information from MRI to direct biopsy needles under real-time US guidance [2,6,7]. Studies have shown that mpMRI combined with TB technology is a promising tool in the diagnosis of PCa [2,8,9]. In light of a number of previous trials showing the significant benefits of image TB, research is now focused on whether random biopsies are required at all in the detection of prostate cancer [10,11,12]. This question is pertinent to settle an ongoing debate as studies have also highlighted that TB with the addition of systematic random biopsy is superior to systematic random biopsy alone either in terms of capturing csPCa or even in terms of post-procedural morbidity [1]. In a large retrospective study, from centres in Europe and the USA, Dell’Oglio et al. aimed at findings a group of men where systematic biopsies could be avoided altogether in men with MRI-facilitated targeted biopsy approach. The authors failed to achieve their objectives and concluded that systematic sampling should be combined with the image-guided fusion biopsies [13]. In a large multicentre prospective study, researchers concluded a higher detection rate for clinically significant prostate cancer for a combined approach (TB and SB) biopsy method; however, different image fusion systems including cognitive guidance were used in the targeted biopsy approach [11]. Furthermore, the study did not use PIRADS v2 categorisation and no standardised protocol was used for MRI imaging. This allowed biases and heterogeneity into the reported data. Our study is a protocol-driven prospective investigation with a standardised US/MRI fusion protocol. We assessed clinical variables that could help in identifying patients who may benefit from systematic random biopsies in addition to fusion targeted approach. The comprehensive analysis and outcomes using methodology of this study has not been reported in the literature [14,15], in particularly the net clinical benefit of the approach.

The aim of this study was to:(1)Compare the diagnostic accuracy of MRI/US fusion targeted biopsies, systematic biopsies and combined approaches in the detection of csPCa and define predictive factors where a combined approach could be used.(2)Quantify additional benefits of adding systematic biopsies to the targeted biopsies approach by constructing a nomogram and assessing its net clinical benefits.

## 2. Materials and Methods

### 2.1. Study Population

The study had ethical approval (14/ES/1070) and all participants provided informed consent for their imaging data to be used. The study also had Caldicott institutional approval through the East of Scotland Ethical committee and Caldicott permission (Caldicott/IGTCAL6358) to link data with electronic system wherever follow-up outcomes were needed. The study period was between April 2015 and March 2020.

The inclusion criteria were age between 40 to 76, abnormal digital rectal examination (DRE), PSA ≤ 20 ng/mL and MRI < T3 disease. Exclusion criteria were repeat biopsies, prior radiotherapy to prostate and diagnosis of acute prostatitis with the last 12 months or a history of PCa. All participants had pre-biopsy mpMRI and only MRI positive (PIRADS ≥ 3) were recruited into the study (*n* = 198). Patients then underwent prostate biopsy by the MRI/US fusion technique (Hitachi HI-RVS; Europe Holding, Steinhausen, Switzerland) by operator 1. This was followed by a standard 12-core TRUS biopsy by a second operator (blinded to the MRI results). In total, 78/198 (39%) underwent radical prostatectomy (RP).

### 2.2. Outcomes

The primary outcome was to compare detection rates of csPCa using both SB and TB approaches alone and in combination. This was assessed both at biopsy and RP stages. csPCa was defined as the presence of prostate cancer with Gleason Score ≥ 3 + 4 (International Society for Urological Pathology [ISUP] grade 2 or more).

The secondary outcome was to assess the net clinical benefit of the approaches using nomogram and decision-analysis methods.

### 2.3. Sample Size Estimation

In considering performing a McNemar matched test, a minimum sample size of 110 men undergoing both SB and TB prostate biopsy approaches would be required to yield 90% power with a significance level of 0.05 (α = 0.05), which would also allow 20% of the dropout rate. The csPCa detection rate via Sb and TB were found in a previous study [16]. Therefore, we recruited more patients than the minimum required number from the sample estimation to ensure achievement of study power and significance level.

### 2.4. Multi-Parametric MRI

All mpMRI scans were performed using 3T scanner (TIM Trio, Siemens, Erlangen, Germany) approximately 2 weeks before TRUS and MRI/US fusion biopsies. The mpMRI protocol was derived from the European Society of Uro-radiology Guidelines 2012 for the detection of prostate cancer and the subsequent publication of version 2 (Appendix A). Briefly, Appendix A summarises the MRI acquisition parameters. Prostate images were acquired in all three imaging planes, whereby the plane of the prostate was defined in relation to the rectal wall.

The mpMRI images were analysed and scored by experienced uro-radiologists (with more than 5 years post-certification experience using PIRADS v2.0). PIRADS v2.0 assessment categories were described as follows: score 1, clinically significant cancer is highly unlikely to be present; score 2, clinically significant cancer is unlikely to be present; score 3, the presence of clinically significant cancer is equivocal; score 4, clinically significant cancer is likely to be present; and score 5, clinically significant cancer is highly likely to be present.

### 2.5. Biopsy Procedures

All mpMRI scans were prepared by an experienced uro-radiologist (MSB) and fusion-targeted biopsies were performed by an experienced radiologist or in their presence using the Hitachi HI-RVS platform (Europe Holding, Steinhausen, Switzerland) using a pre-recorded lesion location. Three cores of tissue were obtained in TB approach from previously identified mpMRI lesions using a superimposed T2-weighted sequence on the real-time TRUS image. The systematic random 12-core biopsies were performed by an experienced urologist or specialist nurse following targeting.

The systematic random biopsy was typically a 12-core approach, collected in an extended sextant template of biopsies from the lateral and medial aspects of the base, mid, and apical prostate from the left and right sides. The biopsy results were analysed by experienced uro-pathologists who were blinded to the MRI findings. The Gleason Score for each patient was obtained.

A subset of the cohort (*n* = 78) underwent RP and their pathological stages were as follows: T2a = 2, T2c = 47, T3a = 24 and T3b = 5. Figure 1 summarises the study protocol in brief. The radical prostate specimens for histology were sliced in patient-specific moulds to aid orientation between imaging and histology per lesion, which were then fabricated using a 3D printer, as described previously by our group and others [17]. Specifically, patient-specific 3D printed moulds were made prior to surgery based on the T2-weighted MRI prostate capsule. The moulds were then customised for each patient using MIMICS and Solidworks. This was used as a reference standard to assess the diagnostic accuracy of both SB and TB in detecting csPCa.

### 2.6. Statistical Analysis

Patients’ age, PSA, prostate volume (mL) and PSAD were collected. PSAD was calculated using PSA divided by the MRI-derived prostate volume (ellipsoid method). The number of MRI lesions, index lesion size (mm), PIRADS category and lesion location were measured by mpMRI. Each lesion was counted only once. The index lesion size was the size of the lesion with the highest PIRADS score. Continuous data were first tested to see if they were normally distributed by the Kolmogorov–Smirnov Test of Normality. The mean (m) and standard deviation (SD) were described if the variable followed a normal distribution. The median (M) and interquartile range (IQR) were presented if the variable was not normally distributed. Categorical variables are reported as frequencies and proportions. Cross tabulation was carried out in order to compare the proportions of csPCa patients by SB, TB, and combined SB + TB. The McNemar chi-square test was conducted in patients who were given both diagnostic tests. McNemar chi-square, degree of freedom (df) and *p*-value were calculated and presented. A two-step logistic regression was performed to identify explanatory variables of csPCa. First, patients’ age, PSAD, lesion size, PIRADS and number of lesions were individually put into a univariate logistic regression model, where the outcome was defined as having csPCa or not. Statistically significant variables were then put into the multivariate logistic regression model. Odds ratio (OR), 95% confidence interval (95% CI) of odds ratio and *p* values were recorded. A nomogram was created based on the statistically significant variables in the final model. The discriminative ability of the predictive model was tested by receiver operating characteristics (ROC) curve and the concordance statistic (c-statistic) was presented. The predicted probabilities of csPCa were plotted against observed probabilities to test the calibration of the model. Decision curve analysis was applied to determine the benefit of the nomogram. In the subgroup analysis, prostatectomies were performed in a group of 78 patients (112 lesions). The detection rate of true significant prostate cancer lesions via SB, TB and combined SB+TB was compared using McNemar chi-square test. Statistical analyses were conducted by SPSS V23.0 and R V4.0.3. The Bonferroni adjustment, which adjusted *p* value by times of tests, was used accounting for multiple testing. The alpha level (adjusted *p*-value) was set at 0.05/times of tests to determine two-tailed significance for McNemar chi-square test.

## 3. Results

### 3.1. Patient Characteristics

The participating patients’ demographic data are shown in in Table 1. A total of 198 patients who underwent both systematic random and TB in the same setting were recruited into the study. Several clinical variables included baseline information (age, PSA level (ng/mL), PSA density (ng/mL^2^) and prostate volume (mL)), multi-parametric magnetic resonance imaging features (number of lesions seen on MRI, index lesion size (mm)) and PIRADS score.

### 3.2. Comparison of the Detection Rate of csPCa between SB, TB and Combined Approaches

The detection rate of csPCa using random biopsy was 51.0% (101/198) and using targeted biopsy was 56.1% (111/198). This was not statistically significant (McNemar chi-square test was χ^2^ = 2.273, df = 1, *p* = 0.132, Odds ratio (OR) = 0.63 (95% CI, 0.34 to 1.16). The results are shown in Figure 2. There were 17 patients (17/198; 8.5%) where the TB approach alone missed csPCa (eight from the same site and nine from normal-looking prostate on MRI). There were 84 patients (84/198; 42.4%) where the positive cores on systematic sampling and TB detected csPCa (72 from the same sector of index lesion and 12 from different sectors away from index lesion). When the TB is negative (69/198; 34.8%), the SB detected clinically insignificant cancer in 12 patients (12/69; 17.3%) and detected csPCa in eight patients (8/69; 11.6%). Twenty-seven (27/198; 13.6%) men were upgraded to csPCa based on TB, while 17 patients (17/198; 8.5%) were upgraded based on SB (χ^2^ = 2.27, *p* = 0.13).

The detection rate of csPCa was higher using combined biopsy (64.6%; 128/198) in comparison to TB (56.1%; 111/198). The McNemar chi-square test result with the Yates correction was statistically significant (χ^2^ = 15.06, df = 1, *p* < 0.001). There was an 8.5% increase in significant prostate cancer detection rate at the patient level using combined biopsy methods compared to using TB alone.

We further validated findings using a subset of the cohort, where the histopathology of RP was used as a reference standard (Figure 3). There were 170 csPCa (170/190; 89.4%) seen on RP histopathology using mould-based approach and counting each focus of cancer. In total, 112 were targeted using MRI/US image fusion method. The TB approach detected 70 of these (70/112; 62.5%), whereas the SB approach detected 54 (54/112; 48.2%). The difference was statistically significant (the McNemar chi-square test result with the Yates correction was χ^2^ = 6.618, df = 1, *p* = 0.010, OR = 0.36 (95% CI, 0.17 to 0.77)). The combined approach to 112 lesions detected more cancers than SB or TB alone (79/112; 70.5%). Compared to SB, the combined approach detected 22.3% more cancers (70.5% vs. 48.2%). The McNemar chi-square test result with the Yates correction was statistically significant (χ^2^ = 23.04, df = 1, *p* < 0.001). Similarly, the combined approach detected 8% more cancers in comparison to TB (70.5% vs. 62.5%). The McNemar chi-square test result with the Yates correction was statistically significant (χ^2^ = 7.111, df = 1, *p* = 0.008.)

Interestingly, there were 11 cancers (11/190; 5.8%) which were labelled as clinically insignificant and all were upgraded to clinically significant using the TB approach. In comparison, there were 24 (24/190; 12.6%) cancers labelled as clinically insignificant and 20 (20/190; 10.5%) were upgraded using the SB approach. The McNemar chi-square test result with the Yates correction was χ^2^ = 0.450, df = 1, *p* = 0.502, OR = 1.50 (95% CI, 0.61 to 3.67). This was not statistically significant.

### 3.3. Univariate and Multivariate Logistic Regression Analysis and Developed Nomogram

In univariate logistic regression, patient’s age, PSAD, Index lesion size and PIRADS were all significant predictors of csPCa detected by random biopsy (Table 2) and were, therefore, inputted into the multivariate analysis. A 6% increase in odds of csPCa by random biopsy was associated with each one-year increase in age (OR = 1.06, 95% CI 1.01–1.12). A PSAD increase of 1 ng/mL was associated with an almost 26-fold increase in odds of csPCa (OR = 25.63, 95% CI 1.93–341.27). Having PIRADS-5 was another significant predictor of csPCa using random biopsy, which was associated with a six-fold increase in odds compared to those with PIRADS-3 (OR = 5.94, 95% CI 1.77–19.93).

The statistically significant independent variables from the multivariate logistic regression model (age, PSAD and PIRADS) were used to develop a nomogram to predict the probability of csPCa using SB (Figure 4). The model demonstrated good discriminative ability (C-statistic = 0.779, 95% CI 0.714–0.843 (Appendix A)).

The calibration plot demonstrated a good agreement between the model predictions and actual observations for detecting csPCa via SB (Figure 5).

### 3.4. Decision Curve Analysis

The outcomes of the decision analysis curve are shown in Figure 6. The net benefit of performing SB in addition to TB on all cases is depicted by the grey line, whereas the black line represents the net benefit of not performing SB (only TB performed). To avoid the harm of unnecessarily intervening on the patients who are disease free and over intervening in the patients with disease, the net benefit of performing SB in addition to TB based on our prediction model with a reasonable range of threshold probabilities is shown as a red line in Figure 6. The net benefit of using our prediction model is to identify patients at risk of having csPCa who will benefit from SB in addition to TB. Our nomogram increased the overall net clinical benefit when the threshold probability was <60% and improved the diagnosis of csPCa compared to avoiding SB biopsy in all.

## 4. Discussion

### 4.1. Principal Findings of the Study

This study assessed detection rate of csPCa using image fusion targeting, random systematic sampling and combination approaches. Patient-based analyses were further validated using lesion-based data from RP histology. There were statistically significant higher detection rates for the combined biopsy approach in comparison to SB or TB alone. The TB approach alone would have missed 17 csPCas. Therefore, the combined approach detected more csPCa than either SB or TB alone. These results are similar to those reported by Filson et al. [18]; however, they were different to those reported in the PRECISION trial [1]. Therefore, in our observations, omitting concurrent SB during image-fusion may run the risk of missing csPCas in around 8.5% of patients. Similar to Cash et al. [19], we observed TB missing a small number csPCa in targeted areas. It is essential that we balance the advantages of concurrent sampling of the prostate during targeting against the risk of side effects and increased detection of clinically insignificant prostate cancers. Avoiding or adding systematic random biopsy at the time of TB remains a challenge for physicians, as knowledge and evidence of decision-making contributing factors still remains known [20]. Our outcomes from the nomogram indicated the excellent advantage (C-index 78% vs. 70%) of using a multivariable prediction model adjusting for clinical and radiological features (age, PSAD and PIRADS). The nomogram could be used to assist in selecting a group of men where a combined biopsy approach would be more useful.

We have also observed no significant advantage of improved characterisation of csPCa using the TB approach as all cancers labelled as clinically insignificant were upgraded on the final histopathology of RP. The challenge of upgrading or under grading would continue with both biopsy approaches as seen in our previous study [21]. There could be various reasons, such as inadequate sampling due to cancer heterogeneity and poor visibility of cribriform architecture on MRI and in biopsies [22].

### 4.2. Study Findings in Context of the Reported Literature

Several retrospective studies have assessed the outcomes of SB in addition to TB for the detection of csPCa. Sathianathen [23] et al. reported a nomogram with C-index = 70%. This nomogram was based on the clinical variables (biopsy naïve, previous biopsy and active surveillance patients) and imaging variables (number of MRI lesions and PIRADS score). The model provided a higher net clinical benefit at a threshold probability of <30%. The model was meant to predict csPCa in systematic random cores only (when TB was negative); however, our findings focused on predicting those patients who will benefit from performing systematic random biopsy in addition to TB (irrespective of target biopsy being positive or negative). Additionally, unlike their study, our nomogram, along with age, PSAD and PIRADS, found that adding these clinical variables to a model yielded a higher C-index (78% vs. 70%). In contrast to the present study, Sathianathen et al. [23] did not report on a validation cohort using RP as a reference standard. Furthermore, and similar to our study, others have reported the possibility of missing significant cancers if the image fusion targeted approach was offered alone [3,24,25,26]. Dell’Oglio et al. [13] failed to identify patients who might benefit from TB alone; therefore, they supported a combination of TB and SB as the preferred approach. In their study, there was no attempt to predict and assess the clinical variables that could help in identifying patients who might derive a greater benefit from systematic random biopsies. Lastly, Falagario et al. [27], highlighted that smaller lesions in big prostates are more likely to be missed in TB biopsies; therefore, they developed a nomogram based on MRI volumetric parameters and clinical information for deciding when SB should be performed in addition to TB. In their study, all patients underwent biparamtric MRI; however, in our analysis, we followed the standard mpMRI using PIRADS [28]. Moreover, the study was a multi-institutional retrospective data analysis of two previously published prospective trials with predominant fusion biopsies being cognitive rather than image-fusion using software. All men in the MULTI-IMPROD study [29,30] had transrectal systematic biopsies; therefore, these trials were not appropriate in answering the research question of the present study. In contrast to this study, however, Falagario et al. [27] provided a range of probabilities of men missing clinically significant cancers, if SB was to be avoided altogether. We reported a set of measurable imaging criteria which could predict the likely benefit of adding SB to TB.

To our knowledge, this study is the first where lesion-based analyses were carried out using mould-based approach for a comprehensive pathological analysis. This confirmed that most csPCas were detected using a combined biopsy approach. csPCas were still missed by biopsies, which may be due to smaller lesions or the cribriform pattern seen on histopathology [31].

### 4.3. Clinical Implications of the Study Findings and Limitation of the Study

Decision-making using critical analysis, especially in situations of uncertainty, cost pressures and likely patient morbidity, is inevitably based on evidence or on a set of observations. In this study, we presented a decision-curve analysis estimating the net clinical benefits of offering a diagnostic test (combined approach to biopsy) in comparison to TB or SB approaches alone. The clinical and radiological observations were used to construct a nomogram, which is then the basis of a decision-making curve. The curve includes intervention for all and intervention for none and provides a background to facilitate discussions with patients. A balance has to be achieved between maximising detection of csPCas and avoiding detection of clinically insignificant cancers.

At present, various nomograms are used mainly for taking into consideration clinical factors, such as age, pre-operative PSA level and PIRADS score of the suspicious cancers. The present study reports a nomogram based on clinical parameters (age, PSAD and PIRADS). The nomogram clearly showed an improved prediction rate, which can be used to perform additional biopsies and the findings have substantial implications for clinicians and researchers in this area. We envisage that this and further research should bring us closer to precise decision-making. There will remain a group of men where systematic the random biopsy approach would bring value in addition to the TB approach, and thereby, improve informed decision-making in the management of men suspected of having prostate cancer.

However, there are some limitations to this study. This is a single-centre study with dedicated uro-radiologist and pathologists. We wanted to explore the association between lesion location in prostate and csPCa via SB but due to low numbers of lesions in TZ, which was not possible. It was not possible to use PIRADS v2.1, since the enrolment to study started before its publication, and this is a similar challenge to any other study published recently on this topic [11]. The nomogram in the present study has been internally validated (cross-validation and bootstrapping). External validation of the nomogram was not carried out in this study, as this would require further prospective multi-centre recruitment of a cohort to test external validity.

## 5. Conclusions

The study reports a nomogram using clinical variables which can assist decision-making during counselling. Patients could be directed towards having systematic sampling of the prostate in addition to an image fusion biopsy approach. The decision analysis curve confirmed a higher net clinical benefit of a combined biopsy approach compared to targeted or random sampling at an acceptable threshold.

## Figures and Tables

**Figure 1 cancers-14-00001-f001:**
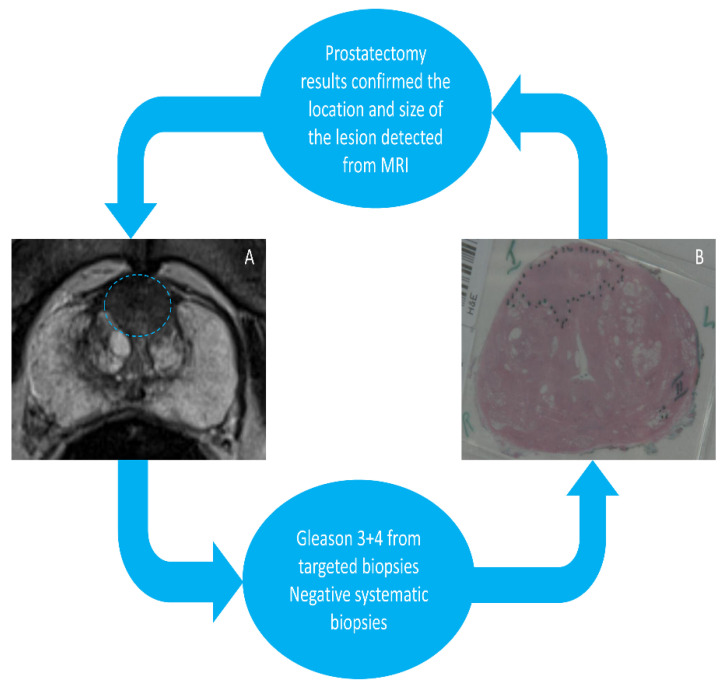
(**A**) A 76-year-old patient with a PIRADS 5 lesion detected from 3T MRI in anterior zone with a high PSA and abnormal DRE. (**B**) Patient-specific 3D mould-based grossing of a radical prostatectomy slice shows a 3 + 4 GS cancer located in the anterior zone.

**Figure 2 cancers-14-00001-f002:**
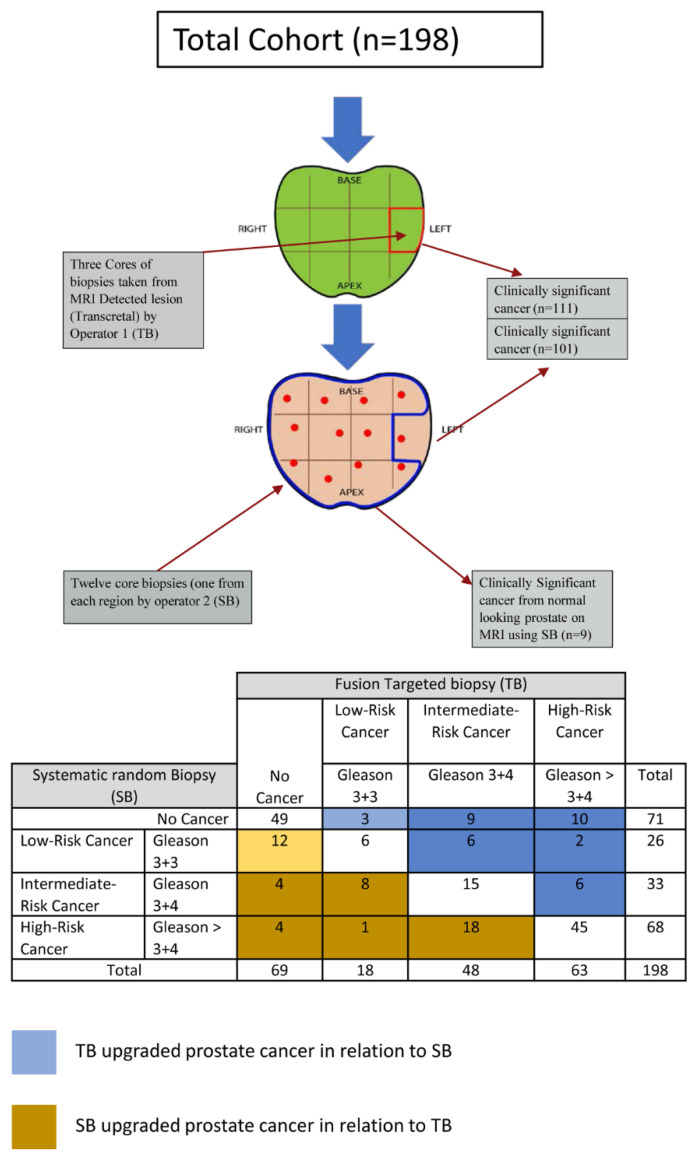
The detection rate of significant prostate cancer between SB and TB based on patients’ level.

**Figure 3 cancers-14-00001-f003:**
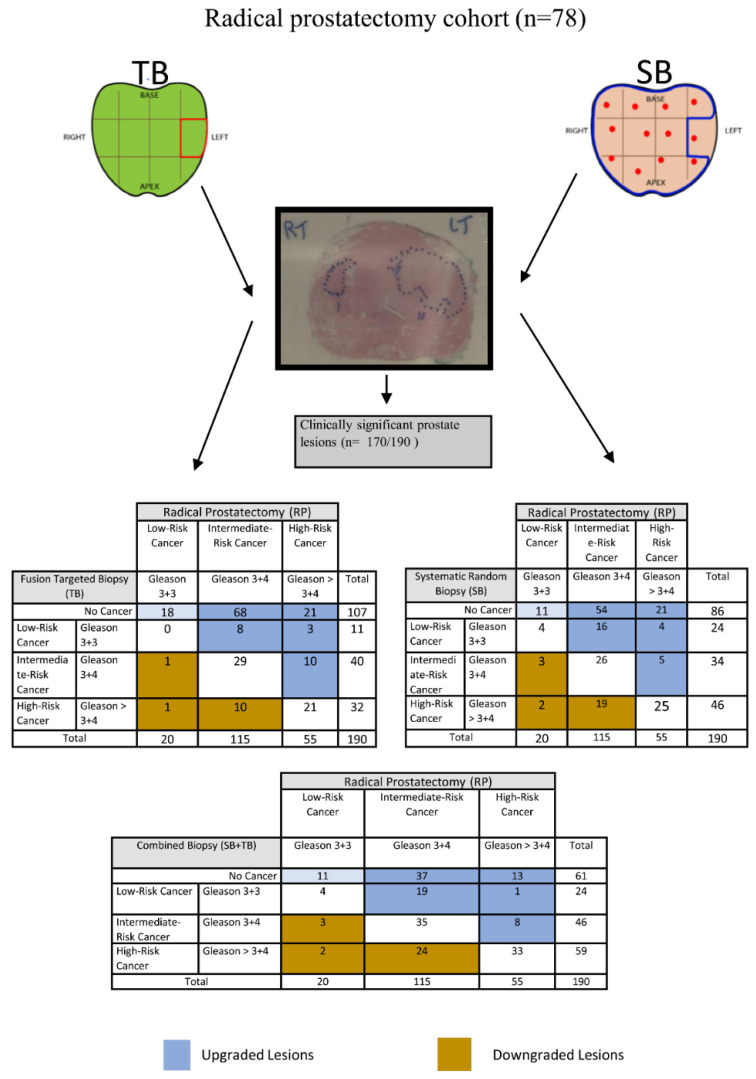
The detection rate of significant prostate cancer via SB, TB and combined SB+TB on RP lesions.

**Figure 4 cancers-14-00001-f004:**
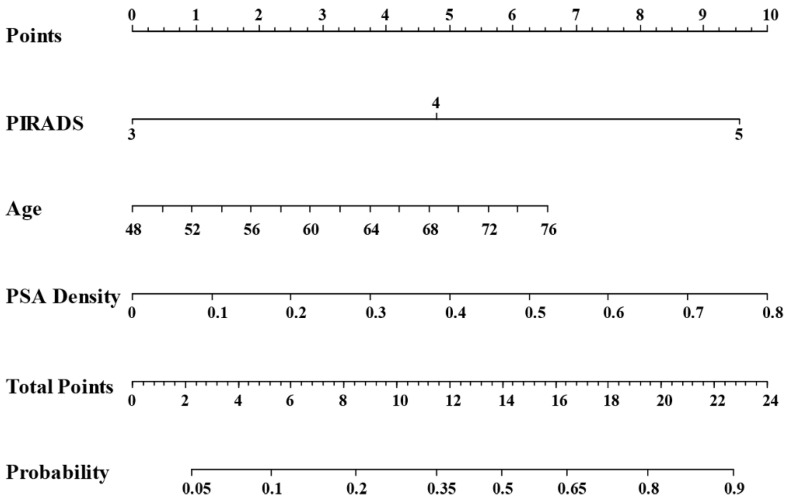
Nomogram with significant clinical variables to predict patients who will benefit from performing systematic random biopsy in addition to TB.

**Figure 5 cancers-14-00001-f005:**
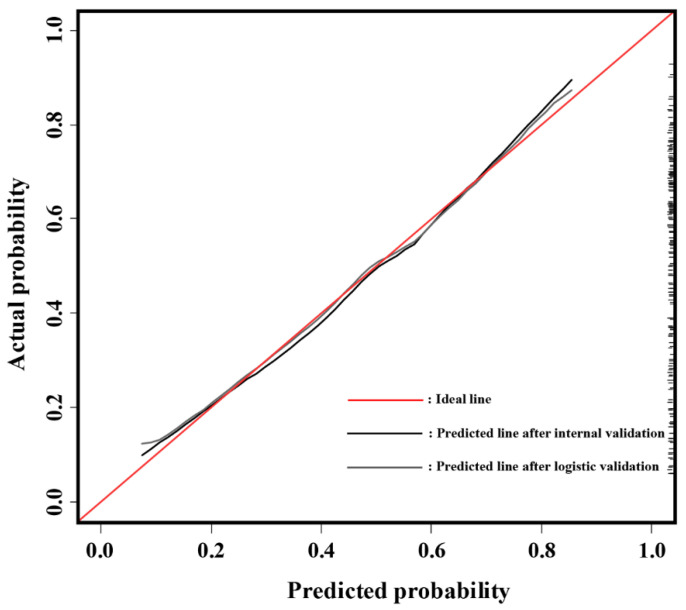
Model calibration plot for observed and predicted probability.

**Figure 6 cancers-14-00001-f006:**
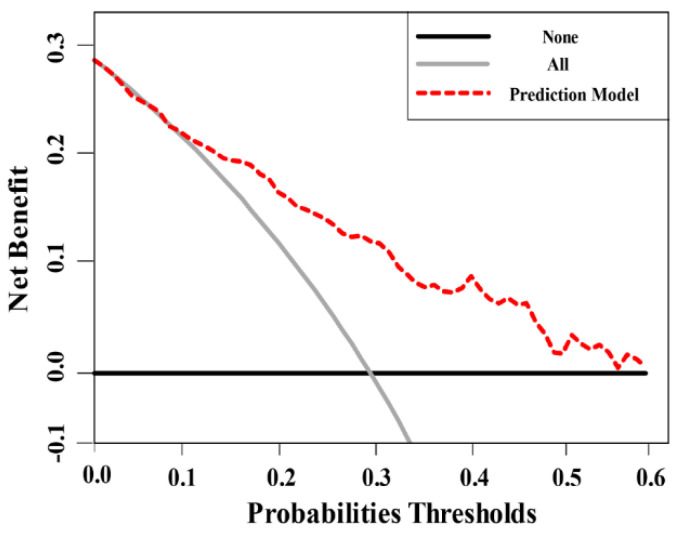
Decision analysis demonstrated a high net benefit of the model across a wide range of threshold probabilities.

**Table 1 cancers-14-00001-t001:** Characteristics of participating patients.

Variables		Overall (*n* = 198)
Basic information	Age, median (IQR), in years	67 (71–61)
	Prostate specific antigen (PSA), median (IQR), ng/mL	8.2 (10.6–6.4)
	Prostate volume, median (IQR), mL	47 (63–33)
	PSA Density, median (IQR), ng/mL^2^	0.18 (0.27–0.11)
mp-MRI	Number of lesions, *n* (%)	
	1	102 (51.5%)
	2	75 (38%)
	3	14 (7%)
	4	6 (3%)
	5	1 (0.5%)
	Index lesion size, median (IQR), mm	16 (25–13)
	Prostate Imaging Reporting and Data System (PIRADS score), *n* (%)	
	PIRADS 3	22 (11%)
	PIRADS 4	55 (28%)
	PIRADS 5	121 (61%)
	Lesion location, *n* (%)	
	Peripheral zone (PZ)	79 (40%)
	Transition zone (TZ)	44 (22%)
	Both zones (TZ-PZ)	75 (38%)
Targeted (TB)/Systematic random (SB) biopsy	Detection of prostate cancer in TB, *n* (%)	129 (65%)
	Detection of prostate cancer in SB, *n* (%)	127 (64%)

**Table 2 cancers-14-00001-t002:** Univariate and multivariate logistic regression analysis.

Covariate	N	Univariate Logistic Regression				Multivariate Logistic Regression			
		OR	95% CI		*p* Value	OR	95% CI		*p* Value
			Lower	Upper			Lower	Upper	
Age (year)	198	1.07	1.02	1.12	0.009	1.06	1.01	1.12	0.036
PSAD	198	92.79	7.61	1130.69	<0.001	25.63	1.93	341.27	0.014
Index lesion size	198	1.06	1.03	1.10	<0.001	1.02	0.98	1.06	0.399
PIRADS	198				<0.001				0.001
3	22	Ref	-		-	Ref	-		-
4	55	1.69	0.49	5.80	0.406	1.51	0.42	5.43	0.525
5	121	9.46	3.00	29.84	<0.001	5.94	1.77	19.93	0.004
Number of Lesions					0.309				
1	102	Ref	-		-				
2	75	1.11	0.61	2.02	0.730				
3 and above	21	2.16	0.81	5.80	0.125				

## Data Availability

The data are available for scrutiny from external requests.

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
