# Peer review of "Predicting the Performance of Concurrent Systematic Random Biopsies during Image Fusion Targeted Sampling of Multi-Parametric MRI Detected Prostate Cancer. A Prospective Study (PRESET Study)"

_cancers, 2021, doi:10.3390/cancers14010001_

Round 1
Reviewer 1 Report
This study was reported the prospective study about the efficacy of combined systematic and target biopsy in patients who had suspicious prostate cancer. The reviewer would like to suggest some critiques as follows.
- On line 48, this sentence is unclear. What is poorly defined? “with, or without systematic…” is correct?
- Additionally, “Drost et al. in a … remains unclear [5]” is difficult to understand for the reviewer, on line 49. The authors should revise this sentence.
- To date, the reviewer thinks that the institution for adopting target biopsy using cognitive method is very rare. Therefore, the authors should revise or delete this sentence, on line 63.
- Likewise, many radiologists seem to adopt PIRADS ver2.1 to diagnose the suspicious of prostate cancer.
- On line 117, the reviewer thinks grammatical error.
- On line 392, reference #4 is wrong.
Author Response
This study was reported the prospective study about the efficacy of combined systematic and target biopsy in patients who had suspicious prostate cancer. The reviewer would like to suggest some critiques as follows.
1/On line 48, this sentence is unclear. What is poorly defined? “with, or without systematic…” is correct?
Ans: The sentence has been revised
2/Additionally, “Drost et al. in a … remains unclear [5]” is difficult to understand for the reviewer, on line 49. The authors should revise this sentence.
Ans: The sentence has been revised to clarify meaning
3/To date, the reviewer thinks that the institution for adopting target biopsy using cognitive method is very rare. Therefore, the authors should revise or delete this sentence, on line 63.
Ans: This may be the case in some institutions, but cognitive method is used by many others. In fact, image fusion method is less used due to cost involved in installing systems.
4/Likewise, many radiologists seem to adopt PIRADS ver2.1 to diagnose the suspicious of prostate cancer.
Ans: Yes, this is the latest version, however our study started much before this version was out and hence data reflects use of previous version.
5/On line 117, the reviewer thinks grammatical error.
Ans: Thank you for the comment. It has been edited. The end bracket has been moved.
6/On line 392, reference #4 is wrong.
Ans: Thank you for the comment. The reference has been modified.

Reviewer 2 Report
Thanks for submitting your paper.
Despite the ubiquity of nomograms in the urologic lierature, there are few nomograms that are helpful and used in real practice. So I would argue that a predictive model should only be published if it is has a compelling clinical use, and that is only rarely the case.
Your model has good c-index and higher net clinical benefit. However, despite your nomogram, in patients with positive pre-biopsy MRI findings, most urologists, including me, will perform SB as well as TB, and such treatment behavior will not change.
Author Response
Thanks for submitting your paper.
Despite the ubiquity of nomograms in the urologic lierature, there are few nomograms that are helpful and used in real practice. So I would argue that a predictive model should only be published if it is has a compelling clinical use, and that is only rarely the case.
Your model has good c-index and higher net clinical benefit. However, despite your nomogram, in patients with positive pre-biopsy MRI findings, most urologists, including me, will perform SB as well as TB, and such treatment behavior will not change.
Ans: We agree with the reviewer, however avoiding or adding additional biopsies to image targeting approach is a hotly debated question in diagnostic pathway of prostate cancer and is a “compelling clinical need” to be answered through research. Hence, we believe that our nomogram would help clinicians in providing further evidence to their practice. The findings of our study are re-assuring to those who are using combined approach including reviewer.

Reviewer 3 Report
Excellent work describing a clinical tool that could be easily accessed by clinicians working with patients with prostate cancer.
Limitations descried - mainly the need for further external validation.
No major changes required in my opinion.
Author Response
Excellent work describing a clinical tool that could be easily accessed by clinicians working with patients with prostate cancer.
Limitations descried - mainly the need for further external validation.
No major changes required in my opinion.
Ans: We have edited limitations section and incorporated suggestions of the reviewer. We appreciate encouraging comments of the reviewer.

Round 2
Reviewer 2 Report
Thank you for your reply.
I agree with you that avoiding or adding additional biopsies to image targeting approach is a hotly debated question in diagnostic pathway of prostate cancer and is a “compelling clinical need” to be answered through research. However, in order for your nomogram to help clinicians in providing further evidence to their practice, your nomogram should be developed based on a larger number of patients and external validation should be done.
Author Response
I agree with you that avoiding or adding additional biopsies to image targeting approach is a hotly debated question in diagnostic pathway of prostate cancer and is a “compelling clinical need” to be answered through research. However, in order for your nomogram to help clinicians in providing further evidence to their practice, your nomogram should be developed based on a larger number of patients and external validation should be done.
Ans:
Thank you for the comment. We agree with the reviewer for a bigger study, despite our study recruited more patients than the minimum required number from the sample estimation to ensure power and significance level. The limitations have been accepted including the need for external validation in the discussion section.
